# Promoting School Engagement in Children with Cerebral Palsy: A Narrative Based Program

**DOI:** 10.3390/ijerph16193634

**Published:** 2019-09-27

**Authors:** Armanda Pereira, Pedro Rosário, Sílvia Lopes, Tânia Moreira, Paula Magalhães, José Carlos Núñez, Guillermo Vallejo, Adriana Sampaio

**Affiliations:** 1Department of Applied Psychology, CIPsi, School of Psychology, University of Minho, 4700-032 Braga, Portugal; 2Faculty of Psychology, University of Oviedo, Oviedo, 33003 Asturias, Spain; 3Facultad de Ciencias Sociales y Humanidades, Universidad Politécnica y Artística de Paraguay, Mayor Sebastián Bullo s/n, Asunción 1628, Paraguay; 4Neuropsychophysiology Lab, CIPsi, School of Psychology, University of Minho, 4700-032 Braga, Portugal

**Keywords:** Cerebral Palsy, executive functions, school engagement, self-regulation, intervention program, narrative

## Abstract

This study assessed the efficacy of an educational program focused on the promotion of school engagement in children with Cerebral Palsy. A 9 weeks, narrative-based intervention program, with a pre-post neuropsychological and self-report evaluation, was developed with a dual focus: a self-regulation theoretical model and executive function stimulation. Fifteen children with Cerebral Palsy participated in the study. Results showed a significant main effect of time (F(2.82) = 6.04, *p* = 0.0066, partial η^2^ = 0.30; F(2.82) = 9.91, *p* = 0.0006, partial η^2^ = 0.41; F(2.82) = 26.90, *p* < 0.0001, partial η^2^ = 0.66) in the three dimensions of school engagement. Findings indicate that the program to train self-regulated competences and executive function skills was efficacious in promoting school engagement in children with Cerebral Palsy. Educational implications were discussed.

## 1. Introduction

Cerebral Palsy (CP) has an estimated prevalence of moderate and severe cases of 1.5 to 2.5 children per 1000 live births [1]. Additionally, an increasing rate of children with CP has been observed due to the growing number of premature infants’ survival [2]. Considering these figures, CP can be considered as one of the most common physical childhood disorders with lifelong impact [3,4,5].

CP comprises a group of neurodevelopmental disorders of movement and posture, holding four criteria: (i) presence of a disorder of movement or posture; (ii) consequence of a cerebral abnormality; (iii) early developmental disorder (prenatal, perinatal, or postnatal development of the brain); and (iv) movement impairment stable in time associated to a non-progressive cerebral abnormality [6,7]. As a broad clinical entity, the motor disorder is likely to vary in its features and associated severity [8]. For example, CP motor impairment could be classified according to the topography of the lesion (e.g., hemiplegia—impairment in the upper and lower extremity of one side of the body) and type of the disorder (e.g., spastic—muscle spasticity) [8,9]. Furthermore, motor impairment has frequently been associated with secondary disturbances, such as those in sensation, perception, cognition, communication, and behavior, as well as the presence of epilepsy, disequilibrium [10], and secondary musculoskeletal problems [11]. Importantly, people with CP present a predisposition to show working memory (WM) and executive function (EF) deficits [12,13,14] with a clear impact on activities of daily living (ADL) and on the learning process [14,15]. Specifically, regarding the latter, the risk that children with CP have to present learning disabilities is beyond the presence of cognitive impairment and highly related with EF deficits, since children with a normative cognitive level can present specific learning difficulties (frequently in mathematics and reading) [12].

EFs are conceptualized as a family of complex and high cognitive processes involved in human cognition [16,17,18,19]. Despite the multiplicity of definitions, EFs are commonly defined as a set of attention-regulation skills, responsible for goal-directed behavior, influenced by previous learning, and with repercussions in future actions and plans [19].

Unlike automatic processes, EFs require effort and the use of self-regulation (SR) competences [20,21]. Specifically, following Diamond [20] EFs show two forms as follows: Core EFs (i.e., inhibition interference control, working memory, and cognitive flexibility) and High Order EFs (i.e., reasoning, problem solving, and planning). Literature suggests that EF and SR overlap, which may be explained by the activation of common prefrontal neural networks [19,21]. According to Diamond [20], SR behavior is highly related to the action of inhibitory control, which contributes to the response inhibition, control of attention, and to a functional level of emotional, motivational, and cognitive arousal. Moreover, EFs and SR behaviors are only activated by will-driven processes to achieve goals [19]. In fact, growing evidence suggests that EFs improve school achievement and vice-versa [19,22,23]. The EF skills (e.g., planning, anticipatory behavior) play a relevant role in SR behavior and are, theoretically, associated to School Engagement (SE) (e.g., follow rules; focus on the task; sustain attention). Children’s school trajectories show that, among other things, SR is positively associated with social competence, motivation, and SE [19,24,25]. Furthermore, Bridgman [26] argued that children’s SR skills and SE hold a strong relationship with school readiness and should start being promoted in kindergarten.

SE has been largely studied in recent literature and is considered the lighthouse guiding efforts to address major educational challenges [27,28]. For example, SE has been considered a relevant framework to understand school dropout, disaffection, and low levels of academic achievement [24,29]. Fredricks and colleagues [24] proposed a model of SE that provides a relevant theoretical framework to the present study. According to these authors [24,29], SE is a multidimensional and multifaceted construct comprising three interrelated but distinct dimensions: (i) behavioral engagement (BE); (ii) emotional engagement (EE); and (iii) cognitive engagement (CE). Behavioral engagement reflects the student’s involvement in school and social activities (intra or extracurricular). This SE component entails school attendance, fulfilment of schoolwork, and participation in class, school, and extracurricular activities [30]. Emotional engagement refers to students’ feelings and affective reactions towards school [31,32]. Students’ interest, happiness, and valuing of school activities reflects a positive emotional engagement [32]. Emotional engagement is also related to school identification [31] and feelings of belonging to the school community. Lastly, cognitive engagement refers to students’ efforts, will, and goals directed towards learning [24]. Positive levels of cognitive engagement are associated with the use of SR strategies [24,33]. In fact, student’s cognitive engagement draws on metacognitive strategies to manage time and effort allocated to tasks, which requires cognitive flexibility, persistence, and self-efficacy to cope with problem solving [34].

Children with CP are especially prone to show poor SE. Due to motor limitations, children are likely to struggle to follow school routines and show limited participation in school activities [26,35]. However, CP has been regarded as a mutable disorder, despite being a permanent developmental condition with persistent neurological prejudice [36]. This changeable nature of CP opens an opportunity for educational interventions, in which “the earlier the better” is the rule of the thumb. Extant literature has highlighted the need to run SE promotional programs including the training of EFs and SR competences focused on ADL or on the school context to foster SE and autonomy [19,37,38]. Grounded on these reasons, the present study aims to examine the efficacy of an educational program focused on the promotion of SE in children with CP.

## 2. Materials and Methods

### 2.1. Participants

Eighteen CP Rehabilitation Centers in Portugal were invited to participate in the research project. Five responded positively (response rate of 28%) and, of these, three were randomly selected. The rehabilitation professionals at each center were asked to select children with CP according to the following criteria: (i) age-range between 8 and 12 years; (ii) cognitive performance level above the Medium-Low (Wechsler Intelligence Scale for Children—Third Edition—WISC-III) and EF impairment; and (iii) epilepsy episodes under control.

As of the time of recruitment, none of the potential research participants held a previous formal intellectual evaluation. All the potential participants were assessed on their intellectual ability using the Wechsler Intelligence Scale for Children—Third Edition (WISC-III) [39]. Participants in the pool of potential participants were enrolled in the fourth, fifth, and sixth grades and all are struggling students with specific difficulties in mathematics and Portuguese language. To cope with these difficulties, they have special support from a special educational teacher on a weekly basis. Moreover, these children showed distinct levels of motor competency (e.g., Diplegia, Left Hemiplegia). The latter was not an inclusion criterion and a relevant aspect for this study, so children were not characterized according to topographic motor type and quality of tonus. For this reason, this variable was not considered in the current study.

The 23 children selected by the rehabilitation staff were invited to participate in the program. All the potential participants were informed about all the phases of the project during a presentation in each Rehabilitation Centre. Of these, 15 (8 females) agreed to participate in the program (response rate of 65%) (see Table 1); all filled out an informed consent form. Data of the parents/caregivers who agreed to participate are presented in Table 1. Those who declined the invitation presented reasons related to difficulties in time management, i.e., most of the children had their week completely booked with school classes and tasks, medical appointments, and therapy sessions. Finally, this study was approved by the Social and Human Science ethics subcommittee of the University of Minho (CEICSH 032/2019).

### 2.2. Procedure

This research followed a quasi-experimental design without a control group to assess the efficacy of an educational program focused on the promotion of school engagement in children with Cerebral Palsy. More specifically, we used a one-group pre-post-test design using double pretest [40]. The randomized controlled trial is generally considered to have the highest level of credibility with regard to assessing causality; however, in this study we choose not to randomize the intervention due to ethical and practical considerations (e.g., not deprive the participants of the eventual potential benefits of the intervention program, small number of children with CP available to participate in the research).

Therefore, all 15 children participated in all steps of the program. The two pre-intervention measurements were collected before the beginning of the program and one final measurement at the end of the program: Pretest 1 (week 0), Pretest 2 (week 9), and Posttest (week 18).

The first measurement (Pretest 1) was used to refute some of the possible threats to internal validity, including maturation effects and regression towards the mean, as plausible alternative explanations to establishing causality for observed associations. While the second measurement (Pretest 2) was used as the baseline upon which to determine if a change had occurred. The intervention program lasted 9 weeks (two sessions per week, with a total of 18 intervention sessions). The sessions were conducted by one researcher with experience in delivering school-based education courses on self-regulation (SR), and training in working with CP children. All sessions of the intervention program were video recorded. In addition, because girls commonly show a physical and mental development more accelerated than boys (e.g., [41]), this study of children with CP provides an opportunity to clarify whether the intervention program affects boys and girls differently.

#### 2.2.1. The Incredible Adventures of Anastácio, the Explorer—A Narrative Intervention Program

The intervention program was designed to promote SE in children with CP. This program is based on a story-tool and aimed to promote children’s autonomy for ADL, school trajectories, and their life project. The program was conducted in the rehabilitation center, on the same days of the usual rehabilitation sessions (e.g., occupational therapy), but on a different schedule. This organization aimed to save the family an extra trip to the rehabilitation center. All sessions were conducted by the first author of this study in a room previously prepared for this purpose.

#### 2.2.2. The Story-Tool

The intervention program draws on the narrative: *The Incredible Adventures of Anastácio, the Explorer*, purposefully built for this investigation. This story (http://anastacio-projecto.weebly.com) was designed for children under the age of 12 [33] and tells the story of a boy who went on an adventure in his own house. Participants are invited to read the narrative and discuss the content and the SR strategies embedded in the story. The narrative begins with a package left at Anastácio’s doorstep full of living tools (a map, a magnifying glass, a pencil, and a notepad), accompanied by a challenging message: “Tomorrow morning, after breakfast, go to the old oak. Bring your new tools.” Anastácio is then assigned a mission: find a plant to cure the sick rabbits living in the nearby forest. To overcome this challenge, Anastácio and his friends (e.g., Cucu, the albino squirrel and Zee, the Bee) face many challenges and obstacles. Throughout the adventure, they learn a set of SR strategies on how to adapt and overcome each obstacle. While reading and discussing the story, readers with CP are likely to learn that it is hard to overcome all the fears and barriers that prevent them from leaving their comfort zone, but eventually, with effort and persistence, they may attain the distinct goals they self-set. Why use a narrative as an intervention tool? Bruner [42] advocates that individuals use narratives as a structure to organize their life events. This predisposition to organize and understand experiences through narratives may work as a facilitator to learn a wide set of SR strategies that help respond to ADL challenges and promote children’s sense of autonomy [43]. Through the reading and discussion of the character’s feelings and responses, children are encouraged to experience vicarious learning [44] and discuss divergent thinking strategies to ADL challenges. Participants learn the SR strategies embedded in the story and are encouraged to transfer this knowledge to their daily challenges [45]. The emerging message of this adventure is “In the story, as in life, full stop points do not exist, only commas” [33], i.e., each outcome holds a lesson and a reflection on the SR process followed, and children are encouraged to reflect on and discuss the challenges they face daily. For example, a school or daily life failure may inform children of the need to improve the use of SR in ADL (e.g., personal study; peers’ interactions; progress in the physiotherapy).

#### 2.2.3. Intervention Program

This intervention has the major purpose of enabling children with CP with the SRL tools needed to help them improve their SE. The program aims to train competencies likely to scaffold the learning process of children with CP. For this reason, this intervention was designed to be run in parallel with classes and rehabilitation sessions.

The present intervention program used Anastácio’s story-tool to promote SE through EF stimulation combined with the promotion of SR competences. This methodology intends to create a proximal environment to help discuss the various difficulties children with developmental problems may encounter in their lives.

#### 2.2.4. The SR Model

SR is an active process in which children assume the control and responsibility of their own behaviors, cognitions, and emotions to pursue their goals. The mastery of SR strategies requires a metacognitive, motivational, and behavioral engagement with a task (e.g., math homework; tying shoelaces) [46]. Children who self-regulate their activities actively monitor behavior, choosing the strategies fitted to attain self-set goals [47]. The SR model by Zimmerman [48] presents the learning process involving three independent phases in a cyclical model (forethought, performance or volitional control, and self-reflection). The PLEE model (i.e., Planning, Execution, and Evaluation) [48,49] adds to the cyclical nature of the SR model with a recursive loop: each phase of the process activates a PLEE cycle that runs within. The cyclical nature of the PLEE model informs and affects the subsequent phases. For example, in the planning phase children are expected to plan (e.g., select a timetable as a time management strategy), execute (e.g., design and fill in the timetable), and evaluate (e.g., check if all elements were included). The recursive nature of the PLEE cycle informs the subsequent phase of the process, helping children to fully learn and experience the SR process [43,47,48,49]. The story-tool, *The Incredible Adventures of Anastácio, the Explorer*, is grounded in the PLEE SR model. Moreover, to support the training of each phase of the PLEE model, the story content discussion is driven by the three types of conceptual knowledge: declarative (what is goal setting?), procedural (how do you set a goal?), and conditional (when do you set a goal?) [50].

#### 2.2.5. Session Structure

The program was comprised of 18 sessions, 60 min each. Throughout the program, children read and discussed the content of the six chapters of the story-tool. Each chapter was analyzed in three sessions as follows: in the first session, the chapter was read by the researcher and a reflection on the content and SR was done. With the help of the researcher, children were encouraged to discuss the SR strategies and messages embedded in the narrative. For example, supported by the story episodes, the discussions promoted a goal-directed reflection through intentional inquiry with children (e.g., Anastacio’s fear of heights—Why does Anastácio feel afraid of being in the top of the old oak? Is being afraid of something common? How do you deal with your fears? What strategies can we use to overcome the fear of failure in school?).

In the second and third sessions, the chapter was retold interactively by the participants, mediated by the researcher. The reflections of the previous sessions were recalled. Afterwards, the children did a consolidation task, i.e., an individual/group activity targeting the promotion of SR. At the end of each session, as a take home message, participants were invited to identify ways to apply what they learned to their daily lives, and to create a slogan about what they had learned in the session. Each session had specific topics (see Appendix A) to be developed and followed a protocol with three steps (for a detailed description, please see Appendix B).

#### 2.2.6. Treatment Integrity

To assure the integrity of the implementation of the protocol conducted by the researcher, two measures were used: (1) During the implementation of the program, on a weekly basis, the principle investigator met with the researcher to discuss project issues and adherence to protocol (e.g., use of the narrative); (2) one expert on SRL and CP watched 30% of the sessions. Data from the video observations indicated that the researcher completed 95% of the activities (range 80–95). Results allowed us to conclude that there was a very good treatment fidelity for the sessions of the program.

### 2.3. Measures

SE was assessed with the SE questionnaire by Wang and Holcombe [28], adapted to the CP population for the purposes of this investigation. Two therapists with extensive experience working with children with CP helped review the items of the questionnaire. The SE questionnaire comprises 13 items presented in a Likert-like format of five points (1 = never to 5 = always). In the current research, the questionnaire was filled in by children and parents/caregivers at the beginning and end of the intervention program. Items were focused on the three dimensions of SE. Behavioral engagement was assessed with three items (e.g., For me it is difficult to finish my homework; School is very important to me) (α_children_ = 69); Emotional engagement was assessed with six items (e.g., For me school is very important) (α_children_ = 72).

Due to the theoretical overlapping of SR competences and EF with cognitive engagement, we have chosen a neuropsychological test battery to assess this dimension: The Tower Test from the Delis–Kaplan EF System (D-KEFS) [51]. The Tower Test is a cognitive task in which children were asked to build a tower with disks (following a model). To accomplish this task, children activate inhibition/anticipation functions which are mainly involved in the use of SR strategies [52]. Children worked with five disks, with different sizes, and had to distribute them among three pegs until completing the task. To accomplish this task, children should follow two rules: (1) never place a big disk above a smaller one and (2) never move more than one disk at a time. The test was administered and data were analyzed by two researchers (first and last author) with experience using this instrument. The results provide information about updating and planning competences. The D-KEFS Tower Test internal consistency coefficients by age range from 0.43 to 0.84 [51,53].

### 2.4. Data Analysis

Many approaches to the analysis of repeated measurements have been proposed, however, likelihood-based, mixed-effects models provide appropriate general analytic frameworks to determine whether fixed effects (i.e., the main effects and the effects associated with the time by sex interaction) and random effects (i.e., the effects associated with the intervention program for each participant) will be included in the model [54]. The mixed-effects model repeated measures (MMRM) analysis implemented herein included an unstructured modelling of time and a within-subject error correlation structure. In this study, time is considered as a classification rather than a continuous variable. Following the rejection of an omnibus hypothesis, next we determined the contrasts between the populations in which the means were not equal to zero. To control the family-wise error rate for all possible pairwise comparisons, the Hochberg [55] step-up Bonferroni inequality was applied using the ESTIMATE statement in SAS PROC MIXED and the HOC option in SAS PROC MULTTEST. The dataset was analyzed using MMRM with ML estimation as implemented in SAS Version 15.1 [56] PROC MIXED.

## 3. Results

### 3.1. Preliminary Analyses

Table 2 provides the descriptive statistics for behavioral, emotional, and cognitive engagement across time and for both male and female groups. In general, findings indicate that all engagement data were reasonably normally distributed; in fact, the examination of the skewness and kurtosis statistics indicated that all values were within the range of ±2. In addition, the standard scores z for every measurement of BE, EE, and CE dimensions were in the range of ±3.0, moreover, none showed extreme cases or presented outliers in the data. Therefore, according to the asymmetry and kurtosis values, no significant violation was found, and data was considered suitable for further analysis.

### 3.2. The Mixed-Effects Model Repeated Measures (MMRM) Analysis

In the absence of a theory providing contrasting data, we used a data-driven strategy to move toward a simpler structure by eliminating predictors or (co)variances that did not appear to be related to the dependent variable. Based on this modeling strategy, the selected model includes fixed effects of the sex and time. Due to the absence of evidence of a statistically significant interaction between sex and time, we focused our interpretation on the main effects. Table 3 shows the MMRM SAS results for the fixed effects of the final model fit for each of three dimensions of SE (i.e., BE, EE, and CE). There was a significant main effect of time in the BE, EE, and CE dimensions (F(2.82) = 6.04, *p* = 0.0066, partial η^2^ = 0.30; F(2.82) = 9.91, *p* = 0.0006, partial η^2^ = 0.41; F(2.82) = 26.90, *p* < 0.0001, partial η^2^ = 0.66). Regarding the variable of classification sex, no statistically significant differences in any of the three dimensions of the SE were found. On the other hand, although it is not explicitly shown, our results indicate that there were statistically significant differences in each of the dimensions of SE, both between children and within children.

The next step was to examine whether the change was different across time in BE, EE, and CE. As indicated in Table 4 (top panel), the means of the two pre-intervention measurements are significantly different or marginally different from those of the post-intervention, after applying the Hochberg’s sequentially rejective Bonferroni procedure (t(28) = −2.91, *p* = 0.0208; t(28) = −2.18, *p* = 0.0570). The definition of marginal significance raises some controversy in the statistical field. We have not been able to find any formal criteria in the literature clarifying when a *p* value marginal is too big, so following common practice, a *p* value ranging from *p* = 0.05 to *p* = 0.075 may be considered acceptable. Hochberg’s post hoc tests of within-subjects for EE (see Table 3, central panel) showed that the means of Pretest 1 versus Posttest, and Pretest 2 versus Posttest are significantly different (t(28) = −4.42, *p* = 0.0004; t(28) = −3.30, *p* = 0.0039). Finally, post hoc tests of within-subjects for CE (see Table 3, bottom panel) indicated that the means of Pretest 2 versus Posttest are significantly different (t(28) = −5.40, *p* = 0.0001). Interestingly, we have not found significant differences between the mean scores regarding the three dimensions of SE from Pretest 1 to Pretest 2 (t(28) = −0.88, *p* = 0.3849; t(28) = 0.05, *p* = 0.9640; t(28) = 0.12, *p* = 0.9046). Therefore, after the intervention, each of the dimensions of SE presented scores higher than those prior to the intervention (Pretest 2), and also higher than the maturational trend from Pretest 1 to Pretest 2 would yield. These results reduce the plausibility that threats to internal design validity (e.g., maturation and regression to the mean) are responsible for the observed changes in the variables. Table 4 also shows that the partial η^2^ values for the significant contrasts, after controlling the family-wise error, ranged from 0.26 to 0.69.

## 4. Discussion

“How do you know I’m on an adventure?” I asked curiously (Anastácio).

“Well, well. We are all engaged on an adventure. Whenever we challenge our limits, we are living an adventure. We may not recognize it, but it would be a shame because all the adventures are … an adventure” (Cucu) [33] (p. 22).

The rationale to run this study is grounded on the: (i) predisposition of children with CP to show poor SE; (ii) the challenge launched by Piovesana and colleagues [38] indicating the need to include EF training in the intervention program’s design; and (iii) work by Zelazo and colleagues (2013) [57] highlighting the relevance of investigating SR associated with EF skills and its impact in school. All things considered, *The Incredible Adventures of Anastácio, the Explorer* story-tool was designed with the purpose of promoting SE in children with CP through the training of EF and SR.

Current findings indicate that participants enrolled in the program improved in BE, EE, and CE. Data also show that time has a significant main effect in each of the dimensions of SE, but the variable sex does not show differences in the three SE dimensions analyzed. These results, despite preliminary, are very encouraging and indicate that the Anastácio program could be considered a valuable tool to promote SE among children with CP. To the best of our knowledge there are no publications reporting educational interventions focused on promoting SR competences and SE in children with CP. Still, according to the systematic review of Novak and colleagues [4], educative interventions with the purpose of improving behavior and social skills (e.g., conductive education) have shown low results with this specific population. Current data are consistent with extent research using story tools to promote SRL [48,58,59]. Reasons for the findings of the current study are believed to be twofold: the use of the narrative as a tool to deliver SRL strategies and the intervention group dynamics. The story tells the adventure of a boy struggling to overcome obstacles and attain his goals. For example, setting personal goals, understanding the nature of fear when coping with challenges, the importance of asking for, and accepting, the help offered by others to overcome obstacles, and ways to approach and solve problems, were topics analyzed in the sessions. Participants were invited to discuss the story and reflect on their own behavior. In these sessions, children met and worked with peers with a similar clinical condition, but with distinct symptoms and functioning (e.g., children with motor and communication limitations). Understanding a wide variety of challenges faced by other children with CP, and the actions needed to address them, may have helped each child to cope with their own problems and improve their competences, for example, in ADL challenges. Children were invited to reflect on Anastácio’s responses to the challenges of the adventure compared to their own responses to their daily adventures (e.g., doing the therapy homework; studying at home; interact with their peers). While discussing the story and reflecting on their own behaviors in the sessions, children were encouraged to complete the tasks of the program and to help each other (e.g., participant A15 struggled to write the task assignments and, alternately, participants helped him finish the tasks during the session). Working with other children with CP and discussing their difficulties to cope with ADL challenges is likely to have helped them reflect metacognitively and improve their efforts to attain their own goals. Our findings are consistent with those by Kim and colleagues (2014) [60] which investigated the effects of action observation of physical training on the actions of the observer (i.e., children with CP). Findings showed positive effects on the upper extremity functions in children with CP as a result of having observed physical training.

The communitarian approach developed in the sessions of the current program may have promoted the feelings of belonging and empathy within the group and, consequently, their SE. This result is consistent with the work by Lovitt and colleagues (1999) [61], which highlighted the positive effects of integrating children with special educational needs in the school community. Participants’ high improvement in EE may suggest the need to build safe environments focused on enabling children with CP to develop their potentialities.

Acknowledging these recommendations and considering the changeable nature of CP [36], the Anastácio intervention program intentionally promoted SRL competences and EF. The development of children’s EF is modulated by experience; in fact, the neural circuit is characterized by high plasticity and is likely to change throughout children’s development [19]. Therefore, children with CP should be supported with interventions focused on the training of strategies related with EF processes [22].

Literature has reported that the three dimensions of SE are interrelated and are likely to influence each other on positive loops [62]. Our findings are consistent with prior research stating that the quality of relationships, the approval by peers and teachers, and students’ sense of belonging at school, may contribute to the development of CE, which may lead to an increase in BE [63]. This is likely to be particularly important for students with special needs due to their vulnerability and low engagement [61,64]. However, future research should consider further analyzing the role of these aspects in the SE of children with CP. In fact, the specificity and complexity of some cases require complementary measures and a deep analysis (e.g., case study) to understand which factors may influence the development of learning strategies knowledge and of its use.

These findings stress the need to tailor interventions addressing each child’s own needs. Moreover, the analysis focused on the individual trajectories helped understand the importance of using measures sensitive to CP’s particularities. In fact, each child faces diverse challenges that the self-reports may not be able to capture. Future investigations may consider the need to use complementary instruments sensitive to the particularities of the CP condition. For example, repeated observational measures could provide an accurate picture of each child’s path during the intervention.

Other limitations may have influenced the findings reported and should be acknowledged. For example, SE is a complex process and to ensure the trustworthiness of the data, researchers should consider using observational measures to help capture all the dimensions of SE. Additionally, the number of participants is limited. Notwithstanding, it was a challenge to gather this number in regular sessions. These children have several health problems associated with the CP clinical condition which limits their regular participation in the sessions; in addition, they have busy schedules full with school activities and therapies. Future research should consider delivering this program using an online format to best meet the needs of children with CP. Our findings, despite preliminary and in need of further research, show that the training of SR and EF offered participants various opportunities to reflect on their own behavior in different contexts (e.g., school, home, therapy), to make choices, to assess the outcomes and assume responsibility for them, and, ultimately, to grow. This was a very important experience for all children; parents reported several anecdotal episodes confirming this proposition. However, to become autonomous and responsible, children with CP need the help of the educators that accompany them on a daily basis. Parents, rehabilitation therapists, and teachers, all play an important role in the promotion of the autonomy and SR of children with CP; thus, researchers may consider enrolling these educators in future investigative designs. For example, schools could consider training teachers to identify and meet the educational needs of their students with CP. To achieve this goal, data collection could be extended to parents, therapists, and teachers to capture the transfer of the acquired competences to ADL (e.g., planning daily study, setting suitable goals), and train them on how to improve children’s SRL strategies and EF. Equipping educators with a set of SRL and EF strategies is expected to further extend the impact of interventions on SE, as the end of the Anastácio’s story suggests: “Mom do you think I can do it too [attain self-set goals]?”/“What part of the story do you want to read again, son?” [33].

## 5. Conclusions

Children with CP face adversities in school activities and ADL due to the specific impairments related to this clinical picture. Deficits in EF, and consequently in SR with an impact on children’s SE, need to be promoted and stimulated. Effective, tailored, narrative-based intervention programs focused on the training of strategies and addressing children’s needs are likely to counteract the impact of specific impairments in school and ADL challenges. During this intervention, children were encouraged to analyze ADL, set goals, learn SRL strategies, and reflect metacognitively on divergent solutions. This set of SRL processes aimed to display children’s agency and their autonomy, which is likely to improve their will and skill competences and their overall quality of life. Future studies could consider starting this program at early ages (6–8 years) to help CP children in early stages of their development.

## Figures and Tables

**Table 1 ijerph-16-03634-t001:** Study participants’ characterization.

Characteristics	Participants (*n* = 15)
Age, mean (SD)	10.53 (1.69)
Gender, male (%)	7 (17.9%)
Diplegia (%)	6 (15.4%)
Right Hemiplegia (%)	4 (10.3%)
Left Hemiplegia (%)	2 (5.1%)
Ataxic (%)	3 (7.7%)
Spastic (%)	10 (25.6%)
Diskenetic (%)	2 (5.1%)
Intellectual Profile, mean (SD)—WISC-IIIn *	
FSIQ	85.33 (15.37)
VIQ	97.27 (13.6)
PIQ	79.67 (15.55)
VCI	97.20 (14.02)
POI	81 (15.23)
PSI	82.33 (18.24)
School Level (%)	
Elementary school	13 (33.3%)
Middle school	2 (5.1%)

* Wechsler Intelligence Scale for Children—Third Edition (WISC-III), Full Scale Inteligence Quocient (FSIQ), Verbal Intelligence Quocient (VIQ), Performance Intelligence Quocient (PIQ), Verbal Comprehension Index (VCI), Perception Organization Index (POI), Processing Speed Index (PSI).

**Table 2 ijerph-16-03634-t002:** Descriptive statistics by sex for each of the three dimensions of School Engagement (SE).

Male							Female				
Engagement	Time	*M*	*Sk*	*Kur*	*SD*	*N*	*M*	*Sk*	*Kur*	*SD*	*N*
	0	2.33	0.98	1.01	0.90	7	1.63	−0.26	−1.97	0.55	8
Behavioral	9	2.43	1.52	2.76	1.12	7	1.83	0.32	0.78	0.64	8
	18	2.38	0.21	−0.23	0.59	7	2.21	0.93	0.06	0.80	8
	0	3.98	−1.13	1.35	0.43	7	4.04	−0.14	−1.80	0.39	8
Emotional	9	3.98	−1.13	1.35	0.43	7	4.04	−0.14	−1.80	0.39	8
	18	4.06	0.16	−1.43	0.56	7	4.39	−1.90	4.41	0.26	8
	0	11.85	−0.23	−0.16	4.30	7	13.87	−0.81	−0.22	5.17	8
Cognitive	9	12.00	1.76	3.92	3.37	7	13.78	−0.01	−0.22	3.58	8
	18	14.29	0.73	0.38	2.69	7	14.50	0.01	0.12	1.92	8

Note: M = Mean; Sk = Skewness; Kur = Kurtosis; SD = Standard Deviation; N = Sample Size.

**Table 3 ijerph-16-03634-t003:** Results of mixed-effects model repeated measures (MMRM) analysis of each of the three dimensions of SE.

	Behavioral Engagement				Emotional Engagement				Cognitive Engagement			
Effect	df_N_	df_D_	F	Pr > F	df_N_	df_D_	F	Pr > F	df_N_	df_D_	F	Pr > F
Sex	1	13	2.82	0.1169	1	13	0.99	0.3391	1	13	0.05	0.8188
Time	2	28	6.04	0.0066	2	28	9.91	0.0006	2	28	26.90	<0.0001

**Table 4 ijerph-16-03634-t004:** Hochberg’s adjusted *p* values and observed size effects for all possible pair-wise differences among the levels of the within-subjects factor.

Pairwise Contrasts for BE
Label	Estimate	SE	df	t-Value	Pr > |t|	Hoc-p	Partial η^2^
Pretest 1 vs. Posttest	−0.4197	0.1440	28	−2.91	0.0208	0.0208	0.395
Pretest 2 vs. Posttest	−0.3703	0.1700	28	−2.18	0.0380	0.0570	0.264
Pretest 1 vs. Pretest 2	−0.0494	0.0550	28	−0.88	0.3849	0.3849	
**Pairwise Contrasts for EE**
**Label**	**Estimate**	**SE**	**df**	**t-Value**	**Pr > |t|**	**Hoc-p**	**Partial η^2^**
Pretest 1 vs. Posttest	−0.2932	0.0663	28	−4.42	<0.0001	0.0004	0.600
Pretest 2 vs. Posttest	−0.2902	0.0879	28	−3.30	0.0026	0.0039	0.458
Pretest 1 vs. Pretest 2	0.0030	0.0663	28	0.05	0.9640	0.9640	
**Pairwise Contrasts for CE**
**Label**	**Estimate**	**SE**	**df**	**t-Value**	**Pr > |t|**	**Hoc-p**	**Partial η^2^**
Pretest 2 vs. Posttest	−2.1626	0.4006	28	−5.40	<0.0001	0.0001	0.692
Pretest 1 vs. Posttest	−1.9812	1.2578	28	−1.59	0.1265	0.2529	
Pretest 1 vs. Pretest 2	0.1814	1.4999	28	0.12	0.9046	0.9046

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
