# Peer review of "Promoting School Engagement in Children with Cerebral Palsy: A Narrative Based Program"

_ijerph, 2019, doi:10.3390/ijerph16193634_

Round 1

Reviewer 1 Report

This is a reasonably good piece of academic work, well planned, well described and worth recommending for publishing.

There are some minor things that could be changed to enhance the quality of the article.

For example in the Introduction section rationale of the study could be enhanced with some paragraphs on CP children educational capacities and achievements and the title suggested I would be reading about school engagement.

Methods and procedure are described well. 

In the Results section table 2 is not necessary as is brings too much data not related directly to the findings of the intervention and may confuse a reader a little, but whether cut it off or leave it in the text I leave up to the authors.

Also table 4 is missing from this section.

Discussion would require in my opinion some contrasting the findings of the authors' intervention method with other methods and intervention carried by other authors.

References are sufficiently up-to-date.

Graphical aspects and editing also fine.

Small comments I have included in the text.  

Author Response

Response to Reviewer 1 Comments

This is a reasonably good piece of academic work, well planned, well described and worth recommending for publishing.

There are some minor things that could be changed to enhance the quality of the article.

Point 1: For example in the Introduction section rationale of the study could be enhanced with some paragraphs on CP children educational capacities and achievements and the title suggested I would be reading about school engagement.

Response 1: Thank you for this comment. We added some paragraphs in the introduction and study purpose sections to address this comment.

Methods and procedure are described well.

Point 2:In the Results section table 2 is not necessary as is brings too much data not related directly to the findings of the intervention and may confuse a reader a little, but whether cut it off or leave it in the text I leave up to the authors.

Response 2:Thank you for this comment. Please, find the answer to this aspect in our response to your comment n.º 7.

Point 3:Also table 4 is missing from this section.

Response 3:Thank you for this comment. Please, find the answer to this comment in our response to your comment n.º 8.

Point 4: Discussion would require in my opinion some contrasting the findings of the authors' intervention method with other methods and intervention carried by other authors.

Response 4:: We thank the reviewer for this comment. We have included a paragraph in the discussion section that could help readers further understand our findings (pag. 10).

Comment 1: I think adding something about educational aspects of children with CP would be a valuable inset for the rationale.

Response comment 1:We thank the reviewer for this comment. In the introduction section, we added information about educational aspects of children with CP that are likely to help readers better understand our rationale (pag. 2).

 Comment 2: Isn't it what attidute ABC in Psychology (affective, bevavioural, cognition) is about?

Response comment 2:This is a very relevant comment and we thank the reviewer for the opportunity to clarify it. We agree that both models show resemblances; however, the focus of each is distinct. Our focus in the current paper is on SE, and that is why we discussed the three factors related to the students' engagement in school.

Comment 3: four years might be a huge gap in the level of skills if the children where treated and trainied properly for motor competency so maybe the authors could add some information on the earlier training experience of the children? Also in terms of educational backgrounds this may be significant.

Response comment 3:We agree with the reviewer that, depending on the issue in analysis,  the gap between the 8 and 12 years could be relevant; and if so, taken into consideration.

We didn’t focus our study in motor competency, this is the reason why this aspect was not considered in the recruitment process of participants or reported in the analytics. For the current study, we only asked that children had a cognitive functioning above the Medium-Low, due to the cognitive demands of the program.  However, the diversity of motor competency helped children better understand that each show distinct difficulties to cope with ADL, and the discussion of why and how we all need help from others, was very relevant to promote children with CP engagement in the program. This idea was included in the discussion section.

However, and to avoid misunderstandings, we deleted the detailed information regarding the topographic motor type and quality of tonus of the participants and explained the reason in the manuscript.

Comment 4: When problem-solving is it possible for the children to arrrive each with different solution or there is just one option of the final solution?

Response comment 4:We thank very much this comment and the attentive reading. We changed the sentence in the method section to clarify the procedure followed (pag 5).

 Comment 5: This is valuable practice to measure long-time effects.

Response comment 5: We truly appreciate the comment.

Comment 6:Did the authors have a specialist on their team to implement the test and then analyse them?

Response comment 6: Thank you, for the opportunity. Two of the authors are trained to implement and analyze the results from Tower Test. This information was included in the manuscript.

Comment 7:I am not sure whether table 2 is necessery for this analyses of the results. I think the paragraph stating the values of discriptive statusitics would be enough. This data doen' t say much about the intervention but it brings chaos of too much data into result analysis instead of focusing on the results.

Response comment 7:Thank you for this suggestion; we agree that sometimes statistical information could not help readers; but in this particular situation, we believe that it is important to keep the table 2, for the sake of clarity, and, for example, to justify our hypothesis.

Comment 8:Table 4 is missing. Is there any reason for that?

Response comment 8: We thank the comment and we are sorry that you didn’t receive information regarding Table 4. All the tables were uploaded in the platform at the same time, and are available in the platform for consultation; we do not know the reason why Table 4 is missing in the manuscript. Notwithstanding, we included Table 4 in the results section of the new version.

Comment 9:Can you contrast your results of the narrative story intervention against other forms of inteventions with similar age CP children?

Response comment 9: We thank the reviewer for this comment. We have included a paragraph regarding this aspect in our discussion section (pag.10).

Comment 10: To minimize or to counteract? What would be the potential benfi of such intervention programme introduced earlier, for example at the age of 6-8?

Response comment 10: Thank you for the latter suggestion that better suits our purpose. We elaborated on the topic and changed the text accordingly.

Reviewer 2 Report

The principal aim of this study was to investigate the efficacy of an educational program focused on the promotion of school engagement in children with cerebral palsy. The results showed a significant main effect of time in three dimensions of school engagement. The authors suggested that the program to train self-regulated competences and executive function skills was efficacious in promoting school engagement in children with cerebral palsy. The study is well-designed and well-conducted, and the results are interesting and substantial for the community. However, authors must rework these few points to refine their article.

Comments:

Abstract

Lines 22-23: Authors should report the results of their analysis at the end of the following sentence: ‘Results showed a significant main effect ……school engagement’. Here are the results to be integrated [F(2,28) = 6.04, p = .0066; F(2,28) = 9.91, p = .0006; F(2,28) = 26.90, p <.0001].

1.1. The study purpose

Lines 54-59: According to Diamond 2013 there are two forms of executive functions, namely: Core EFs (Inhibition, working memory and cognitive flexibility) and Higher-order EFs (reasoning, problem solving and planning). I suggest that the authors complete this part with a short paragraph that clearly explains the forms of executive functions and then clearly specify that they are the forms of executive functions that are in relation to SR behavior.

Materials and methods

I would like to know if the protocol has been validated by an ethics committee? if so, specify it in the method section.

Line 114: In table 1, participants do not have the same level according to the GMFCS. I would like to know how the authors controlled this bias since this bias was not considered in the statistical analyses.

Results

Line 279: I suggest to the authors to add the effect sizes

Line 317: Put the table 4

Author Response

Response to Reviewer 2 Comments

The principal aim of this study was to investigate the efficacy of an educational program focused on the promotion of school engagement in children with cerebral palsy. The results showed a significant main effect of time in three dimensions of school engagement.

The authors suggested that the program to train self-regulated competences and executive function skills was efficacious in promoting school engagement in children with cerebral palsy. The study is well-designed and well-conducted, and the results are interesting and substantial for the community. However, authors must rework these few points to refine their article.

Comments

 Point 1

Abstract

Lines 22-23: Authors should report the results of their analysis at the end of the following sentence: ‘Results showed a significant main effect ……school engagement’. Here are the results to be integrated [F(2,28) = 6.04, p = .0066; F(2,28) = 9.91, p = .0006; F(2,28) = 26.90, p <.0001].

Response 1: We thank the reviewer for this comment. We have included this information in our abstract section (line: 22-24).

Methods and procedure are described well.

Point 2

The study purpose

Lines 54-59: According to Diamond 2013 there are two forms of executive functions, namely: Core EFs (Inhibition, working memory and cognitive flexibility) and Higher-order EFs (reasoning, problem solving and planning). I suggest that the authors complete this part with a short paragraph that clearly explains the forms of executive functions and then clearly specify that they are the forms of executive functions that are in relation to SR behavior.

Response 2:We thank the reviewer for this comment. We have included a short paragraph as suggested in our study purpose section (pag 2).

Point 3: I would like to know if the protocol has been validated by an ethics committee? if so, specify it in the method section.

Response 3:Thank you for this comment. Information regarding the ethics committee was further detailed in the method section.

Point 4

Line 114: In table 1, participants do not have the same level according to the GMFCS. I would like to know how the authors controlled this bias since this bias was not considered in the statistical analyses.

Response 4:: We thank the reviewer for this relevant comment.

We didn’t focus our study in motor competency, this is the reason why this aspect was not considered in the recruitment process of participants or reported in the analytics. For the current study, we only asked that children had a cognitive functioning above the Medium-Low, due to the cognitive demands of the program. However, the diversity of motor competency helped children better understand that each show distinct difficulties to cope with ADL, and the discussion of why and how we all need help from others, was very relevant to promote children with CP engagement in the program. This idea was included in the discussion section.

Still, and to avoid misunderstandings, we deleted the detailed information regarding the topographic motor type and quality of tonus of the participants and explained the reason in the manuscript.

Point 5:  Line 279: I suggest to the authors to add the effect sizes

Response 5:: We thank the reviewer this suggestion. We included this information in the abstract and result section (pag. 1 and 7)

Point 6: Line 317: Put the table 4

Response 6:Thank you. We are sorry that you didn’t receive information regarding Table 4. All the tables were uploaded in the platform at the same time, and are available in the platform for consultation; we do not know the reason why Table 4 is missing in the manuscript. Notwithstanding, we included Table 4 in the results section of the new version.